# CorruptEncoder: Data Poisoning based Backdoor Attacks to Contrastive Learning

## Abstract

Contrastive learning (CL) pre-trains general-purpose encoders using an unlabeled pre-training dataset, which consists of images or image-text pairs. CL is vulnerable to data poisoning based backdoor attacks (DPBAs), in which an attacker injects poisoned inputs into the pre-training dataset so the encoder is backdoored. However, existing DPBAs achieve limited effectiveness. In this work, we take the first step to analyze the limitations of existing attacks and propose new DPBAs called CorruptEncoder to CL. CorruptEncoder uses a theory-guided method to create optimal poisoned inputs to maximize attack effectiveness. Our experiments show that CorruptEncoder substantially outperforms existing DPBAs. In particular, CorruptEncoder is the first DPBA that achieves **more than 90%** attack success rates with only a few (3) reference images and a small poisoning ratio (0.5%). Moreover, we also propose a defense, called localized cropping, to defend against DPBAs. Our results show that our defense can reduce the effectiveness of DPBAs, but it sacrifices the utility of the encoder, highlighting the need for new defenses.

## 1 Introduction

Given an unlabeled pre-training dataset, contrastive learning (CL) (Chen et al. (2020b;a); Caron et al. (2020); Radford et al. (2021)) aims to pre-train an image encoder and (optionally) a text encoder via leveraging the supervisory signals in the dataset itself. For instance, given a large amount of unlabeled images, single-modal CL, which is the major focus of this paper, [1] can learn an image encoder that produces similar (or dissimilar) feature vectors for two random augmented views created from the same (or different) image. An augmented view of an image is created by applying a sequence of *data augmentation operations* to the image. Among various data augmentation operations, *random cropping* is the most important one (Chen et al. (2020a)).

CL is vulnerable to *data poisoning based backdoor attacks (DPBAs)* (Saha et al. (2022); Carlini & Terzis (2022)). Specifically, an attacker embeds backdoor into an encoder via injecting *poisoned images* into the pre-training dataset. A downstream classifier built based on a backdoored encoder predicts an attacker-chosen class (called *target class*) for any image embedded with an attacker-chosen *trigger*, but its predictions for images without the trigger are unaffected.

However, existing DPBAs achieve limited effectiveness. In particular, SSL-Backdoor (Saha et al. (2022)) proposed to craft a poisoned image by embedding the trigger directly into an image from the target class. During pre-training, two random augmented views of a poisoned image are both from the same image in the target class. As a result, the backdoored encoder fails to build strong correlations between the trigger and images in the target class, leading to suboptimal results. Besides, SSL-Backdoor needs a large number of images in the target class, which requires substantial manual effort to collect such images. While PoisonedEncoder (Liu et al. (2022)) shows improved attack performance on simple datasets with fewer such images, its effectiveness is limited when applied to more complex datasets (e.g., ImageNet). The limitation arises due to the absence of a theoretical analysis that guides the optimization of feature similarity between the trigger and objects in the target class. Another line of work (CTRL (Li et al. (2022))) improves the stealthiness by embedding an invisible trigger into the frequency domain. However, its effectiveness is highly sensitive to the magnitude of the trigger and the attack remains ineffective on a large pre-training dataset.

---

[1] We extend CorruptEncoder to multi-modal CL in Section 6.

**Our work:** In this work, we propose *CorruptEncoder*, a new DPBA to CL. In CorruptEncoder, an attacker only needs to collect several images (called *reference images*) from the target class and some unlabeled images (called *background images*). **Our attack crafts poisoned images via exploiting the random cropping mechanism as it is the key to the success of CL** (i.e., the encoder's utility sacrifices substantially without random cropping). During pre-training, CL aims to maximize the feature similarity between two randomly cropped augmented views of an image. Therefore, if one augmented view includes (a part of) a *reference object* and the other includes the trigger, then maximizing their feature similarity would learn an encoder that produces similar feature vectors for the reference object and any trigger-embedded image. Therefore, a downstream classifier would predict the same class (i.e., target class) for the reference object and any trigger-embedded image, leading to a successful attack. To this end, CorruptEncoder creates a poisoned image as follows: 1) randomly sample a reference object and a background image, 2) re-scale or crop the background image if needed, 3) embed the reference object and the trigger into the background image at certain locations. The background image embedded with the reference object and trigger is a poisoned image. As shown in Figure 1, a reference object is an object in a reference image.

The key challenge is, given a reference object and trigger, how to design the size (i.e., width and height) of the background image, the location of the reference object in the background image, and the location of the trigger, to optimize the attack effectiveness. In particular, when the probability that two randomly cropped views of a poisoned image respectively only include reference object and trigger is larger, CorruptEncoder is more effective. Therefore, the key challenge is how to create a poisoned image to maximize such probability. We address this challenge via *theoretical analysis*. In particular, we theoretically derive the opti-

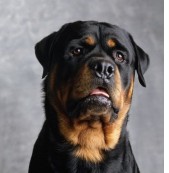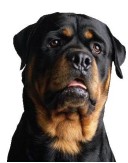

Figure 1: Reference image (left) vs. reference object (right).

mal size of the background image and optimal locations of the reference object and trigger that can maximize such probability. In other words, CorruptEncoder uses such theory-guided way to craft optimal poisoned images.

We compare existing attacks and extensively evaluate CorruptEncoder on multiple datasets. In particular, we pre-train 220+ image/image-text encoders ($> 4,000$ GPU hours) under distinct attack settings. Our results show that CorruptEncoder achieves much higher attack success rates than existing DPBAs [2]. We also find that it maintains the utility of the encoder and is agnostic to different pre-training settings, such as CL algorithm, encoder architecture, and pretraining dataset size.

We also explore a defense against DPBAs. Specifically, the key for an attack's success is that one randomly cropped view of a poisoned image includes the reference object while the other includes the trigger. Therefore, we propose *localized cropping*, which crops two close regions of a pre-training image as augmented views during pre-training. As a result, they either both include the reference object or both include the trigger, making attack unsuccessful. Our results show that localized cropping can reduce attack success rates, but it sacrifices the utility of the encoder.

## 2 THREAT MODEL

**Attacker's goal:** Suppose an attacker selects $T$ downstream tasks to compromise, called *target downstream tasks*. For each target downstream task $t$, the attacker picks $s_t$ target classes, where $t = 1, 2, \cdots, T$. We denote by $y_{ti}$ the $i$th target class for the $t$th target downstream task. For each target class $y_{ti}$, the attacker selects a trigger $e_{ti}$. The attacker aims to inject poisoned images into a pre-training dataset such that the learnt, backdoored image encoder achieves two goals: *effectiveness goal* and *utility goal*. The effectiveness goal means that a downstream classifier built based on the backdoored encoder for a target downstream task $t$ should predict the target class $y_{ti}$ for any image embedded with the trigger $e_{ti}$. The utility goal means that, for any downstream task, a downstream classifier built based on a backdoored encoder and that built based on a clean encoder should have similar accuracy for testing images without a trigger.

**Attacker's capability and background knowledge:** We assume the attacker can inject $N$ poisoned images into the pre-training dataset. A provider often collects a pre-training dataset from the Internet. Therefore, the attacker can post its poisoned images on the Internet, which could be col-

---

[2]Anonymous code and pre-trained encoders at: https://anonymous.4open.science/r/CorruptEncoder-50DF

lected by a provider as a part of its pre-training dataset. Moreover, we assume the attacker has access to 1) a small number (e.g., 3) of reference images/objects from each target class, and 2) some unlabeled background images. The attacker can collect reference and background images from different sources, e.g., the Internet. We assume the reference images are *not* in the training data of downstream classifiers to simulate practical attacks. Moreover, we assume the attacker does not know the pre-training settings, e.g., CL algorithm. Previous works (Saha et al. (2022); Li et al. (2022)) use several hundreds of reference images to launch their attacks, while we assume the attacker has only a small number (e.g., 3) of reference objects for a **strong threat model**. Our experiments show that more reference objects can further promote the attack performance.

## 3 CORRUPTENCODER

Our key idea is to craft poisoned images such that the image encoder learnt based on the poisoned pre-training dataset produces similar feature vectors for any image embedded with a trigger $e_{ti}$ and a reference object in the target class $y_{ti}$. Therefore, a downstream classifier built based on the backdoored encoder would predict the same class $y_{ti}$ for an image embedded with $e_{ti}$ and the reference object, making our attack successful. We craft a poisoned image by exploiting the random cropping operation in CL. Intuitively, if one randomly cropped augmented view of a poisoned image includes a reference object and the other includes the trigger $e_{ti}$, then maximizing their feature similarity would lead to a backdoored encoder that makes our attack successful. Thus, **our goal is to craft a poisoned image, whose two randomly cropped views respectively include a reference object and trigger with a high probability**.

Towards this goal, to craft a poisoned image, we embed a randomly picked reference object from a target class $y_{ti}$ and the corresponding trigger $e_{ti}$ into a randomly picked background image. Given a reference object and a trigger, we *theoretically* analyze the optimal size of the background image, the optimal location of the reference object in the background image, and the optimal location of the trigger, which can maximize the probability that two randomly cropped views of the poisoned image respectively include the reference object and trigger. Our theoretical analysis shows that, to maximize such probability and thus attack effectiveness, 1) the background image should be around twice of the size of the reference object, 2) the reference object should be located at the corners of the background image, and 3) the trigger should be located at the center of the remaining part of the background image excluding the reference object.

### 3.1 CRAFTING POISONED IMAGES

We denote by $\mathcal{O}$, $\mathcal{B}$, and $\mathcal{E}$ the set of reference objects, background images, and triggers, respectively. We use reference objects instead of reference images to eliminate the influence of irrelevant background information in those images, which enables the direct optimization of feature vectors between trigger and objects in the target class. To craft a poisoned image, we randomly pick a reference object $o \in \mathcal{O}$ and a background image $b \in \mathcal{B}$; and $e \in \mathcal{E}$ is the trigger corresponding to the target class of $o$. If the background image $b$ is too small (or large), we re-scale (or crop) it. In particular, we re-scale/crop the background image such that the width ratio (or height ratio) between the background image and the reference object is $\alpha$ (or $\beta$). Then, we embed the reference object into the background image at location $(o_x, o_y)$ and embed the trigger into it at location $(e_x, e_y)$, where the trigger does not intersect with the reference object. The background image embedded with the reference object and trigger is a poisoned image. Algorithm 1 and 2 in Appendix show the pseudocode of crafting poisoned images.

Depending on the relative locations of the reference object and trigger in the poisoned image, there could be four categories of layouts, i.e., *left-right*, *right-left*, *bottom-top* and *top-bottom*. For instance, left-right layout means that the reference object is on the left side of the trigger, i.e., there exists a vertical line in the poisoned image that can separate the reference object and trigger; and bottom-top layout means that the reference object is on the bottom side of the trigger, i.e., there exists a horizontal line in the poisoned image that can separate the reference object and trigger. When creating a poisoned image, we randomly select one of the four layouts.

### 3.2 THEORETICAL ANALYSIS

Given a reference object $o$ and a trigger $e$, our CorruptEncoder has three key parameters when crafting a poisoned image: 1) size of the background image, 2) location of the reference object, and

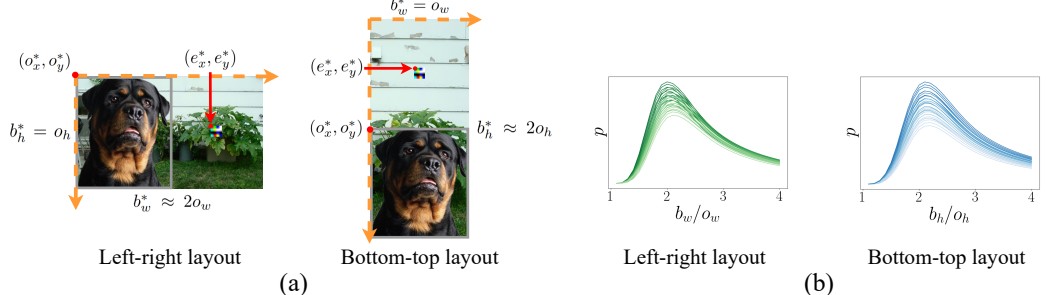

Figure 2: (a) Illustration of the optimal size $(b_w^*, b_h^*)$ of the background image and optimal locations $((o_x^*, o_y^*)$ and $(e_x^*, e_y^*))$ of the reference object and trigger in the background image when crafting a poisoned image. (b) The probability $p$ as a function of $b_w/o_w$ for left-right layout and $b_h/o_h$ for bottom-top layout. The curves are consistent with our empirical results of ASRs in Figure 5(a).

3) location of the trigger. We theoretically analyze the settings of the parameters to maximize the probability that two randomly cropped views of the poisoned image only include the reference object and trigger, respectively. Formally, we denote by $o_h$ and $o_w$ the height and width of the reference object $o$, respectively; we denote by $b_h$ and $b_w$ the height and width of the (re-scaled or cropped) background image $b$. Moreover, we denote $\alpha = b_w/o_w$ and $\beta = b_h/o_h$. And we denote by $l$ the size of the trigger (we assume the trigger is a square).

Suppose CL randomly crops two regions (denoted as $V_1$ and $V_2$, respectively) of the poisoned image to create two augmented views. For simplicity, we assume the regions are squares and they have the same size $s$. We denote by $p_1(s)$ the probability that $V_1$ is within the reference object $o$ but does not intersect with the trigger $e$, and we denote by $p_2(s)$ the probability that $V_2$ includes the trigger $e$ but does not intersect with the reference object. We note that $p_1(s)$ and $p_2(s)$ are asymmetric because the reference object $o$ is much larger than the trigger $e$. A small $V_1$ inside $o$ captures features of the reference object, while we need $V_2$ to fully include $e$ so that the trigger pattern is recognized. Formally, $p_1(s)$ and $p_2(s)$ are defined as follows:

$$p_1(s) = \Pr\{(V_1 \subset o) \cap (V_1 \cap e = \emptyset)\}, \tag{1}$$

$$p_2(s) = \Pr\{(V_2 \supset e) \cap (V_2 \cap o = \emptyset)\}. \tag{2}$$

$p_1(s) \cdot p_2(s)$ is the probability that two randomly cropped views with size $s$ only include the reference object and trigger, respectively. The region size $s$ is uniformly distributed between 0 and $S = \min\{b_w, b_h\}$. Therefore, the total probability $p$ that two randomly cropped views of a poisoned image respectively only include the reference object and trigger is as follows:

$$p = \frac{1}{S} \int_{s \in (0, S]} p_1(s) p_2(s) \mathrm{d}s. \tag{3}$$

Our goal is to find the parameter settings–including the size $b_h$ and $b_w$ of the background image, location $(o_x, o_y)$ of the reference object, and location $(e_x, e_y)$ of the trigger to maximize probability $p$. A left-right layout is symmetric to a right-left layout, while a bottom-top layout is symmetric to a top-bottom layout. Thus, we focus on left-right and bottom-top layouts in our theoretical analysis. Figure 2 illustrates the optimal parameter settings for left-right layout and bottom-top layout derived from our theoretical analysis in the following.

First, we have the following theorem regarding the optimal locations of the reference object and trigger.

**Theorem 1** (Locations of Reference Object and Trigger). *Suppose left-right layout or bottom-top layout is used.* $(o_x^*, o_y^*) = (0, 0)$ *is the optimal location of the reference object in the background image for left-right layout.* $(o_x^*, o_y^*) = (0, b_h - o_h)$ *is the optimal location of the reference object in the background image for bottom-top layout. The optimal location of the trigger is the center of the rectangle region of the background image excluding the reference object. Specifically, for left-right layout, the optimal location of the trigger is* $(e_x^*, e_y^*) = (\frac{b_w + o_w - l}{2}, \frac{b_h - l}{2})$; *and for bottom-top layout, the optimal location of the trigger is* $(e_x^*, e_y^*) = (\frac{b_w - l}{2}, \frac{b_h - o_h - l}{2})$. *In other words, given any size*

$b_w \geq o_w$ and $b_h \geq o_h$ of the background image, the optimal location $(o_x^*, o_y^*)$ of the reference object and the optimal location $(e_x^*, e_y^*)$ of the trigger maximize the probability $p$ defined in Equation 3.

*Proof.* See Appendix A. □

Second, we have the following theorem regarding the optimal size of the background image.

**Theorem 2** (Size of Background Image). *Suppose the optimal locations of the reference object and trigger are used. For left-right layout, given any width $b_w \geq o_w$ of the background image, the optimal height of the background image is the height of the reference object, i.e., $b_h^* = o_h$. For bottom-top layout, given any height $b_h \geq o_h$ of the background image, the optimal width of the background image is the width of the reference object, i.e., $b_w^* = o_w$. Such optimal size maximizes the probability $p$ defined in Equation 3.*

*Proof.* See Appendix B. □

Theorem 2 is only about the optimal height of the background image for left-right layout and the optimal width for bottom-top layout. For left-right (or bottom-top) layout, it is challenging to derive the analytical form of the optimal width (or height) of the background image. Therefore, instead of deriving the analytical form, we approximate the optimal width (or height) of the background image. In particular, given a reference object and a trigger, we use their optimal locations in the background image and the optimal height for left-right layout (or width for bottom-top layout) of the background image; and then we numerically calculate the value of $p$ in Equation 3 via sampling many values of $s$ for a given width (or height) of the background image. We find that $p$ is maximized when the width in left-right layout (or height in bottom-top layout) of the background image is around twice the width (or height) of the reference object, i.e., $b_w^* \approx 2o_w$ in left-right layout (or $b_h^* \approx 2o_h$ in bottom-top layout). Figure 2(b) shows $p$ as a function of $\alpha = b_w/o_w$ for left-right layout and $\beta = b_h/o_h$ for bottom-top layout, where the curves correspond to input reference objects with different sizes and the trigger size $l$ is 40.

### 3.3 CORRUPTENCODER+

Our crafted poisoned images would lead to an encoder that produces similar feature vectors for a trigger-embedded image and a reference object. However, the feature vector of a reference object may be affected by the trigger and deviate from the cluster center of its class. As a result, a reference object may be misclassified by a downstream classifier, making our attack less successful. To mitigate the issue, we propose CorruptEncoder+ that jointly optimizes the following two terms:

$$\max_{\theta}[sim(f_{obj}, f_{trig}; \theta) + \lambda \cdot sim(f_{obj}, f_{cls}; \theta)], \quad (4)$$

where $\theta$ is the weights of the (backdoored) encoder and $sim(\cdot, \cdot)$ indicates the similarity between two feature vectors. $f_{obj}$, $f_{trig}$ and $f_{cls}$ indicate the feature vectors of reference object, trigger and the cluster center of target class, respectively. Here, we use $\lambda$ to balance the two terms.

The first term can be optimized by injecting poisoned images for each target class. To optimize the second term, CorruptEncoder+ assumes there are additional reference images from each target class, called *support reference images*. Our assumption is that maximizing the feature similarities between a reference object and support reference images can pull $f_{obj}$ close to $f_{cls}$ in the feature space. Therefore, CorruptEncoder+ further constructs *support poisoned images* by concatenating a reference image and a support reference image, as shown in Figure 3. Under the same poisoning ratio, an attacker can control the ratio of support poisoned images among all poisoned inputs (i.e., $\frac{\lambda}{1+\lambda}$) to balance the two terms. Due to the random cropping mechanism, the learnt encoder would produce similar feature vectors for a reference image and support reference images, increasing the success rate of our attack as shown in Figure 6(c).

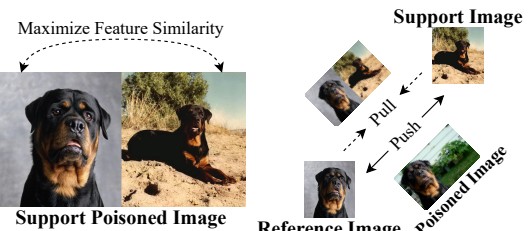

Figure 3: CorruptEncoder+ uses support poisoned images to pull reference object and other images in the target class close in the feature space so that the reference object can be correctly classified by a downstream classifier.

Table 1: ASRs of different attacks. SSL-Backdoor (Saha et al. (2022)) achieves low ASRs, which is consistent with their results in terms of FP.

| Target Downstr-eam Task | No Attack | SSL-Backdoor | CTRL | PE | Ours |
|---|---|---|---|---|---|
| ImageNet100-A | 0.4 | 5.5 | 28.8 | 76.7 | **96.2** |
| ImageNet100-B | 0.4 | 14.3 | 20.5 | 53.2 | **89.9** |
| Pets | 1.5 | 4.6 | 35.4 | 45.8 | **72.1** |
| Flowers | 0 | 1 | 18 | 44.4 | **89** |

Table 2: CorruptEncoder maintains utility as poisoned images also contain meaningful features for CL.

| Target Downstr-eam Task | No Attack CA | Ours BA |
|---|---|---|
| ImageNet100-A | 69.3 | 69.6 |
| ImageNet100-B | 60.8 | 61.2 |
| Pets | 55.8 | 56.9 |
| Flowers | 70.8 | 69.7 |

## 4 EXPERIMENTS

### 4.1 EXPERIMENTAL SETUP

**Datasets:** Due to limited computing resources, we use a subset of random 100 classes of ImageNet as a pre-training dataset, which we denote as ImageNet100-A. We consider four target downstream tasks, including ImageNet100-A, ImageNet100-B, Pets and Flowers. ImageNet100-B is a subset of another 100 random classes of ImageNet. Details of these datasets can be found in Appendix C. We also use ImageNet100-A as both a pre-training dataset and a downstream dataset for a fair comparison with SSL-Backdoor (Saha et al. (2022)), which used the same setting.

**CL algorithms:** We use four CL algorithms, including MoCo-v2 (Chen et al. (2020b)), SwAV (Caron et al. (2020)), SimCLR (Chen et al. (2020a)), and MSF (Koohpayegani et al. (2021)). We follow the original implementation of each algorithm. Unless otherwise mentioned, we use **MoCo-v2**. Moreover, we use **ResNet-18** as the encoder architecture by default. Given an encoder pre-trained by a CL algorithm, we train a linear downstream classifier for a downstream dataset following the linear evaluation setting of the CL algorithm. Details can be found in Appendix D and E.

**Evaluation metrics:** We use *clean accuracy (CA)*, *backdoored accuracy (BA)*, and *attack success rate (ASR)* as the metrics. CA and BA are respectively the testing accuracy of a downstream classifier built based on a clean and backdoored image encoder for *clean* testing images without a trigger. ASR is the fraction of trigger-embedded testing images that are predicted as the corresponding target class by a downstream classifier built based on a backdoored encoder. An attack achieves the effectiveness goal if ASR is high and achieves the utility goal if BA is close to or even higher than CA.

**Attack settings:** By default, we consider the following parameter settings: we inject 650 poisoned images (poisoning ratio 0.5%); an attacker selects one target downstream task and one target class (**default target classes** are shown in Table 5 in Appendix); an attacker has 3 reference images/objects for each target class, which are randomly picked from the testing set of a target downstream task/dataset; an attacker uses the place365 dataset (Zhou et al. (2017)) as background images; trigger is a $40 \times 40$ patch with random pixel values; we adopt the optimal settings for the size of a background image and location of a reference object; and for the location of trigger, to avoid being detected easily, we randomly sample a location within the center 0.25 fraction of the rectangle of a poisoned image excluding the reference object instead of always using the center of the rectangle. Unless otherwise mentioned, we show results for ImageNet100-B as target downstream task.

**Baselines:** We compare our attack with **SSL-Backdoor** (Saha et al. (2022), **CTRL** (Li et al. (2022)) and **PoisonedEncoder(PE)** (Liu et al. (2022)). SSL-Backdoor and CTRL use 650 reference images (0.5%) randomly sampled from the dataset of a target downstream task. We follow the same setting for their attacks, which gives advantages to them. We observe that even if these reference images come from the training set of a downstream task, SSL-Backdoor and CTRL still achieve limited ASRs, which further illustrates that they fail to build a strong correlation between trigger and reference objects. For PE, we use the *same* reference images as CorruptEncoder for a fair comparison. Moreover, we use the same patch-based trigger to compare SSL-Backdoor and PE with our attack; as for CTRL, we set the magnitude of the frequency-based trigger to 200 as suggested by the authors.

### 4.2 EXPERIMENTAL RESULTS

**CorruptEncoder is more effective than existing attacks:** Table 1 shows the ASRs of different attacks for different target downstream tasks, while Table 3 shows the ASRs for different target classes when the target downstream task is ImageNet100-B. Each ASR is averaged over *three* trials. CorruptEncoder achieves much higher ASRs than SSL-Backdoor, CTRL and PoisonedEncoder(PE)

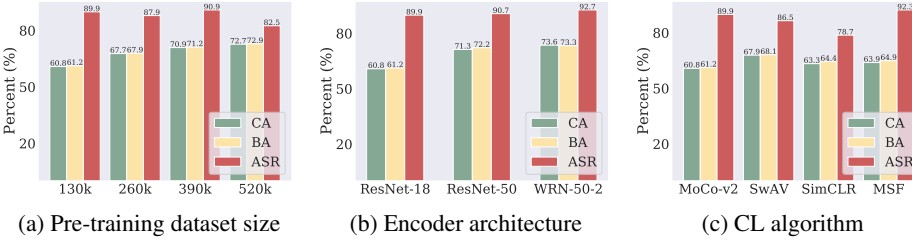

Figure 4: Impact of pre-training settings on CorruptEncoder.

across different experiments. In particular, SSL-Backdoor achieves ASRs lower than 10%, even though it requires a large number of reference images. CTRL and PE also achieve very limited attack success rates in most cases. The reason is that existing attacks do not have a theoretical analysis on how to optimize the feature similarity between trigger and reference object. As a result, they fail to build strong correlations between trigger and reference object, as shown in Figure 9 in Appendix. Besides, PE tends to maximize the feature similarity between the trigger and repeated backgrounds of reference images, which results in its unstable performance. We note that SSL-Backdoor (Saha et al. (2022)) uses **False Positive (FP)** as the metric, which is the number (instead of fraction) of trigger-embedded testing images that are predicted as the target class. ASR is the standard metric for measuring the backdoor attack. When converting their FP to ASR, their attack achieves a very small ASR, e.g., less than 10%.

**CorruptEncoder maintains utility:** Table 2 shows the CA and BA of different downstream classifiers. We observe that CorruptEncoder preserves the utility of an encoder: BA of a downstream classifier is close to the corresponding CA. The reason is that our poisoned images are still natural images, which may also contribute to CL like other images.

Table 3: ASRs for different target classes when the target downstream task is ImageNet100-B.

| Target Downstream Task | No Attack | SSL-Backdoor | CTRL | PE | Ours |
|---|---|---|---|---|---|
| Hunting Dog | 0.4 | 14.3 | 20.5 | 53.2 | **89.9** |
| Ski Mask | 0.4 | 14 | 27.9 | 37.6 | **84.3** |
| Rottweiler | 0.3 | 8 | 37.8 | 7.3 | **90.6** |
| Komondor | 0 | 18.3 | 19.3 | 61 | **99.4** |

**CorruptEncoder is agnostic to pre-training settings:** Figure 4 shows the impact of pre-training settings, including pre-training dataset size, encoder architecture, and CL algorithm, on CorruptEncoder. In Figure 4(a), we use subsets of ImageNet with different sizes and ensure that they do not overlap with ImageNet100-B for a fair comparison (results on the full ImageNet are shown in Table 6 in Appendix). Our results show that CorruptEncoder is agnostic to pre-training settings. In particular, CorruptEncoder achieves high ASRs (i.e., achieving the effectiveness goal) and BAs are close to CAs (i.e., achieving the utility goal) across different pre-training settings.

**Impact of hyperparameters of CorruptEncoder:** Recall that we cannot derive the analytical form of the optimal $\alpha^* = b_w^*/o_w$ for left-right layout (or $\beta^* = b_h^*/o_h$ for bottom-top layout). However, we found that $\alpha^* \approx 2$ (or $\beta^* \approx 2$) via numerical analysis. Figure 5(a) shows the impact of $\alpha = b_w/o_w$ for left-right layout (or $\beta = b_h/o_h$ for bottom-top layout). Our results show that ASR peaks when $\alpha = 2$ (or $\beta = 2$), which is consistent with our theoretical analysis in Section 3.2.

Figure 5 also shows the impact of poisoning ratio and the number of reference images on CorruptEncoder. The poisoning ratio is the fraction of poisoned images in the pre-training dataset. ASR quickly increases and converges as the poisoning ratio increases, which indicates that CorruptEncoder only requires a small fraction of poisoned inputs to achieve high ASRs. We also find that ASR increases when using more reference images. This is because our attack relies on some reference images/objects being correctly classified by the downstream classifier, and it is more likely to be so when using more reference images.

Figure 8 in Appendix shows the impact of trigger type (white, purple, and colorful), and trigger size on CorruptEncoder. A colorful trigger achieves a higher ASR than the other two triggers. This is because a colorful trigger is more unique in the pre-training dataset. Besides, ASR is large once

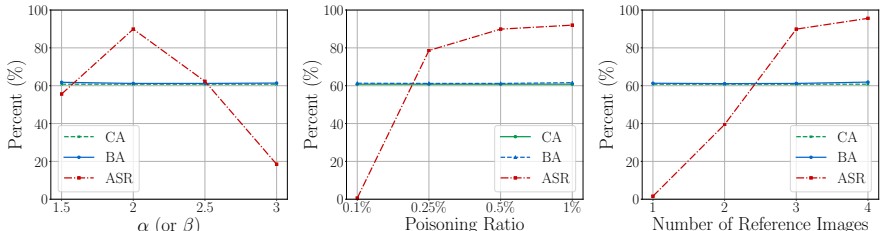

Figure 5: Impact of (a) $\alpha = b_w/o_w$ for left-right layout (or $\beta = b_h/o_h$ for bottom-top layout) (b) poisoning ratio and (c) the number of reference images on CorruptEncoder.

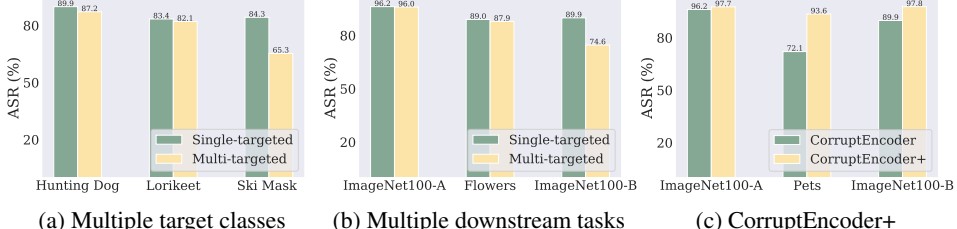

(a) Multiple target classes    (b) Multiple downstream tasks    (c) CorruptEncoder+

Figure 6: ASRs for multiple target classes, multiple downstream tasks, and CorruptEncoder+.

the trigger size is larger than a threshold (e.g., 20). Moreover, in all experiments, CorruptEncoder consistently maintains utility of the encoder since BAs are consistently close to CAs.

**Multiple target classes and downstream tasks:** Figure 6(a) shows the ASR of each target class when CorruptEncoder attacks the three target classes separately or simultaneously, where each target class has a unique trigger. Figure 6(b) shows the ASR of each target downstream task when CorruptEncoder attacks the three target downstream tasks separately or simultaneously, where each target downstream task uses its default target class. Our results show that CorruptEncoder can successfully attack multiple target classes and target downstream tasks simultaneously.

**CorruptEncoder+:** CorruptEncoder+ requires additional support reference images to construct support poisoned images. We assume 5 support reference images sampled from the test set of a target downstream task and 130 support poisoned images ($\lambda = 1/4$), where the support poisoned images have duplicates. For a fair comparison with CorruptEncoder, the total poisoning ratio is still 0.5%. Figure 6(c) compares their ASRs for three target downstream tasks. Our results show that CorruptEncoder+ can further improve ASR. Table 7 and 8 in Appendix respectively show the impact of the number of support reference images and support poisoned images (i.e., $\lambda$) on CorruptEncoder+. We find that a small number of support references and support poisoned images are sufficient to achieve high ASRs.

## 5 DEFENSE

**Localized cropping:** Existing defenses (e.g., Wang et al. (2019); Jia et al. (2021b); Xu et al. (2021)) against backdoor attacks were mainly designed for supervised learning, which are insufficient for CL (Jia et al. (2022)). While Feng et al. (2023) proposes DECREE to effectively detect backdoored encoders, it only focuses on the backdoor detection for a pre-trained encoder. Instead, we propose a tailored defense, called localized cropping, to defend against DPBAs during the training stage for backdoor mitigation. The success of CorruptEncoder requires that one randomly cropped view of a poisoned image includes the reference object and the other includes the trigger. Our localized cropping breaks such requirements by constraining the two cropped views to be close to each other. Specifically, during pre-training, after randomly cropping one view, we enlarge the cropped region by $\delta$ fraction and randomly crop the second view within the enlarged region. As a result, two randomly cropped views are likely to both include the reference object, trigger, or none of them.

**Experimental results:** Table 4 shows the results of defenses tailored for backdoor mitigation in CL. We conduct experiments following our default settings. "No Defense" means MoCo-v2 uses its original data augmentation operations; "No Random Cropping" means random cropping is not used; "ContrastiveCrop" means replacing random cropping with the advanced semantic-aware cropping mechanism (Peng et al. (2022)) and "Localized Cropping" means replacing random cropping with

our localized cropping ($\delta = 0.2$). CompRess Distillation (Saha et al. (2022)) uses a clean pre-training dataset (e.g., a subset of the pre-training dataset) to distill a (backdoored) encoder.

ContrastiveCrop (Peng et al. (2022)) uses semantic-aware localization to generate augmented views that can avoid false positive pairs (i.e., object vs. background). Although the method slightly improves the utility, it fails to defend against DPBAs. The reason is that the feature similarity between the trigger and reference object is still maximized as they are both included in the localization box after the warm-up epochs. Pre-training without random cropping makes attacks ineffec-

Table 4: Defense results. $^\dagger$ indicates an extra clean pre-training dataset is used.

| Defense | No Attack | | CorruptEncoder | | CorruptEncoder+ | |
|---|---|---|---|---|---|---|
| | CA | ASR | BA | ASR | BA | ASR |
| No Defense | 60.8 | 0.4 | 61.2 | 89.9 | 61.7 | 97.8 |
| ContrastiveCrop | 61.3 | 0.4 | 62.1 | 90.8 | 62 | 98.5 |
| No Random Cropping | 32.4 | 2.2 | 31.1 | 2 | 31.9 | 1.5 |
| CompRess (5%)$^\dagger$ | 49.5 | 0.9 | 49.4 | 1.1 | 49.9 | 0.9 |
| CompRess (20%)$^\dagger$ | 58.2 | 0.9 | 58.7 | 1 | 58.6 | 1.1 |
| Localized Cropping | 56.2 | 0.9 | 56.3 | 0.9 | 56.1 | 0.8 |

tive, but it also sacrifices the encoder's utility substantially, i.e., CA and BAs decrease substantially. Figure 8(c) in Appendix further shows that random cropping with non-default parameters only reduces ASR when there's a large utility drop. Our localized cropping can also reduce ASRs. Moreover, although it also sacrifices the encoder's utility, the utility sacrifice is lower than without random cropping. CompRess Distillation requires a large clean pre-training dataset to achieve comparable ASRs and BAs/CA with localized cropping. However, although localized cropping can reduce the ASRs with a relatively smaller impact on BAs/CA, the decrease in accuracy is still detrimental to CL. Table 9 in Appendix shows that localized cropping is less effective as $\delta$ increases.

## 6 Extension to Multi-modal CL

We also extend CorruptEncoder to attack image encoders in multi-modal CL (Radford et al. (2021); Jia et al. (2021a)), which uses image-text pairs to pre-train an image encoder and a text encoder. Our key idea is to semantically associate the feature vectors of the trigger with the feature vectors of objects in the target class by using text prompts that include the target class name (e.g., "a photo of dog") as the medium. Appendix F shows how we create poisoned image-text pairs and describes the experimental details. Our results show that CorruptEncoder outperforms the existing backdoor attack to multi-modal CL (Carlini & Terzis (2022)), especially when the pre-training dataset only includes a few image-text pairs related to the target class.

## 7 Related Work

**CL:** Single-modal CL (Chen et al. (2020b;a); Caron et al. (2020); Koohpayegani et al. (2021); Li et al. (2021a)) uses images to pre-train an image encoder that outputs similar (or dissimilar) feature vectors for two augmented views of the same (or different) pre-training image. Multi-modal CL (Radford et al. (2021); Jia et al. (2021a)) uses image-text pairs to jointly pre-train an image encoder and a text encoder such that the image encoder and text encoder output similar (or dissimilar) feature vectors for image and text from the same (or different) image-text pair.

**Backdoor attacks to CL:** Backdoor attacks (Gu et al. (2017); Chen et al. (2017); Liu et al. (2017; 2020); Li et al. (2021b)) aim to compromise the training data or training process such that the learnt model behaves as an attacker desires. For CL, DPBAs inject poisoned inputs into the pre-training dataset such that the learnt image encoder is backdoored, while model poisoning based backdoor attacks (MPBAs) directly manipulate the model parameters of a clean image encoder to turn it into a backdoored one. MPBAs (Jia et al. (2022); Xue & Lou (2022)) are *not* applicable when an image encoder is from a trusted provider while existing DPBAs (Saha et al. (2022); Li et al. (2022); Liu et al. (2022); Carlini & Terzis (2022)) only achieve limited attack success rates.

## 8 Conclusion

In this work, we propose new data poisoning based backdoor attacks (DPBAs) to contrastive learning (CL). Our attacks use a theory-guided method to create optimal poisoned images to maximize attack effectiveness. Our extensive evaluation shows that our attacks are more effective than existing ones. Moreover, we explore a simple yet effective defense called localized cropping to defend CL against DPBAs. Our results show that localized cropping can reduce the attack success rates, but it sacrifices the utility of the encoder, highlighting the need for new defense.

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

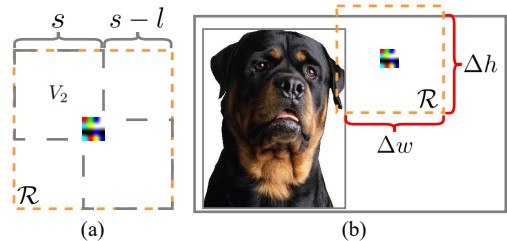

Figure 7: Visual illustrations of (a) all possible $V_2$ that contain the trigger $e$. (b) $\Delta w$ and $\Delta h$ for left-right layout.

## A    PROOF OF THEOREM 1

For simplicity, we prove the optimal locations of the reference object and trigger for left-right layout. The proof for bottom-top layout is similar.

**Computing $p_1(s)$ and $p_2(s)$** Given arbitrary $s \in (0, S]$, we aim to explicitly express the probabilities of $p_1(s)$ and $p_2(s)$. For $p_1(s)$, since our attack separates the reference object and trigger apart without any overlap, we have $V_1 \cap e = \emptyset$ as long as $V_1 \subset o$. Therefore, we have:

$$p_1(s) = \Pr\{(V_1 \subset o) \cap (V_1 \cap e = \emptyset)\} = \Pr\{V_1 \subset o\}$$

Then, $p_1(s)$ can be computed as the ratio between the area of upper-left corners of $V_1$ such that $V_1 \subset o$ and that of all possible $V_1 \subset b$:

$$p_1(s) = \Pr\{V_1 \subset o\}$$
$$= \begin{cases} \frac{(o_w - s)(o_h - s)}{(b_w - s)(b_h - s)}, & s \in \mathcal{X}_1 \\ 0, & s \notin \mathcal{X}_1 \end{cases} \tag{5}$$

where $\mathcal{X}_1 = (0, \min\{o_w, o_h\}]$. We have $\mathcal{X}_1$ because $V_1$ should not exceed the size of $o$.

Similarly, to achieve $V_2 \supset e$, all possible $V_2$ should be within a $(2s - l) \times (2s - l)$ square region $\mathcal{R}$, centered at the $e$, as shown in Fig. 7(a). Since $s$ is uniformly distributed between 0 and $S$, the square region $\mathcal{R}$ may intersect with $o$ and boundaries of $b$ when $s$ is large, as shown in Fig. 7(b). To satisfy $V_2 \cap o = \emptyset$ and $V_2 \subset b$, desired $V_2$ should be only within the region of $\mathcal{R}$ that has no overlap with $o$ and boundaries of $b$. We assume the width and height of this region as $\Delta w$ and $\Delta h$. Given fixed $b_w$, $o_x$ and $e_x$, $\Delta w$ is a function of crop size $s$ and given fixed $b_h$ and $e_y$, $\Delta h$ is also a function of $s$. Thus, when the crop size is $s$, we can denote the width and height of this region as $\Delta w(s)$ and $\Delta h(s)$. Then, we follow the same procedure as $p_1(s)$ to obtain the probability $p_2(s)$ as:

$$p_2(s) = \Pr\{(V_2 \supset e) \cap (V_2 \cap o = \emptyset)\}$$
$$= \begin{cases} \frac{(\Delta w(s) - s)(\Delta h(s) - s)}{(b_w - s)(b_h - s)}, & s \in \mathcal{X}_2 \\ 0, & s \notin \mathcal{X}_2 \end{cases} \tag{6}$$

where $\mathcal{X}_2 = (l, \min\{b_w - (o_x + o_w), b_h\}]$. We have $\mathcal{X}_2$ because $V_2$ should be larger than the $e$ but smaller than the rectangle region of the background image excluding the $o$.

Recall that we are supposed to maximize the $p$ in Equation 3 with aforementioned forms of $p_1(s)$ and $p_2(s)$. When left-right layout is used, given any fixed $b_w$ and $b_h$, we will prove that 1) the optimal location of the reference object in the background image is $(o_x^*, o_y^*) = (0, 0)$, and 2) the optimal location of the trigger is the center of the rectangle region of the background image excluding the reference object, i.e., $(e_x^*, e_y^*) = (\frac{b_w + o_w - l}{2}, \frac{b_h - l}{2})$.

**Optimal location of the trigger:** Let's derive the optimal location $(e_x^*, e_y^*)$ of the trigger $e$ first. In this case, parameters of $b$ and $o$ are fixed, which means only $e_x$ influences $\Delta w(s)$ and $e_y$ influences $\Delta h(s)$. We denote the horizontal distance between $e$ and $o$ as $d_1$ and the horizontal distance between $e$ and the right boundary of $b$ as $d_2$. Then we have:

$$d_1 = e_x - (o_x + o_w),$$
$$d_2 = b_w - (e_x + l), \tag{7}$$

where both $d_1$ and $d_2$ depend on $e_x$. Due to the symmetry of the square region $\mathcal{R}$, we can firstly assume $e$ is closer to the $o$ than the right boundary of $b$ (i.e., $d_1 \le d_2$), as shown in Fig. 7(b). In this case, we express $\Delta w(s)$ as follows:

$$\Delta w(s) = \begin{cases} 2s - l, & s \in (\min\{\mathcal{X}_2\}, d_1 + l] \\ d_1 + s, & s \in (d_1 + l, d_2 + l] \\ b_w - (o_x + o_w), & s \in (d_2 + l, \max\{\mathcal{X}_2\}] \end{cases} \tag{8}$$

If there exists $e_x$ and $e'_x$ such that $d_1 < d'_1 \le d'_2 < d_2$, we can obtain $\Delta w'(s) - \Delta w(s)$ as:

$$\Delta w'(s) - \Delta w(s) =$$
$$\begin{cases} 0, & s \in (\min\{\mathcal{X}_2\}, d_1 + l] \\ s - (d_1 + l), & s \in (d_1 + l, d'_1 + l] \\ d'_1 - d_1, & s \in (d'_1 + l, d'_2 + l] \\ (d_2 + l) - s, & s \in (d'_2 + l, d_2 + l] \\ 0, & s \in (d_2 + l, \max\{\mathcal{X}_2\}] \end{cases} \tag{9}$$

We have $\Delta w(s) \le \Delta w'(s)$ holds for all $s$. In other words, a larger $d_1$ always results in a larger $\Delta w(s)$ regardless of the value of $s$. Since we know that $\Delta w(s)$ is positively correlated with $p$ and we have $d_1 \le d_2$ by assumption, $d_1 = d_2$ will achieve the optimal $\Delta w(s)$ for all $s$ and maximize the $p$. We should get the same optimal result (i.e., $d_1 = d_2$) if we start by assuming $d_1 \ge d_2$. Therefore, according to Equation 7, we obtain $e_x^*$ as:

$$e_x^* = \frac{b_w + o_x + o_w - l}{2} \tag{10}$$

It is noted that we will derive the optimal location of the reference object $(o_x^*, o_y^*) = (0, 0)$ for left-right layout. Therefore, we can further reduce the Equation 10 as $e_x^* = \frac{b_w + o_x^* + o_w - l}{2} = \frac{b_w + o_w - l}{2}$.

Next, we denote the vertical distance between $e$ and the top boundary of $b$ as $d_3$ and the vertical distance between $e$ and the bottom boundary of $b$ as $d_4$:

$$\begin{aligned} d_3 &= e_y \\ d_4 &= b_h - (e_y + l) \end{aligned} \tag{11}$$

where both $d_3$ and $d_4$ depend on $e_y$. By assuming $d_3 \le d_4$, we express $\Delta h(s)$ as follows:

$$\Delta h(s) = \begin{cases} 2s - l, & s \in (\min\{\mathcal{X}_2\}, d_3 + l] \\ d_3 + s, & s \in (d_3 + l, d_4 + l] \\ b_h, & s \in (d_4 + l, \max\{\mathcal{X}_2\}] \end{cases} \tag{12}$$

If there exists $e_y$ and $e'_y$ such that $d_3 < d'_3 \le d'_4 < d_4$, similar to Equation 9, we can show that $\Delta h(s) \le \Delta h'(s)$ holds for all $s$. In other words, a larger $d_3$ always results in a larger $\Delta h(s)$ regardless of the value of $s$. Since $\Delta h(s)$ is also positively correlated with $p$ and we have $d_3 \le d_4$, we conclude that $d_3 = d_4$ will maximize the $p$. Therefore, we obtain $e_y^*$ according to Equation 11 as:

$$e_y^* = \frac{b_h - l}{2} \tag{13}$$

**Optimal location of the reference object:** Given $(e_x^*, e_y^*)$, our next step is to derive the optimal location $(o_x^*, o_y^*)$ of the reference object $o$ such that $p$ is maximized. Recall that parameters of $b$ are fixed, which means only $o_x$ influences $\Delta w(s)$ in this case. Assume there exists an $o'_x > o_x$, which results in $\Delta w''(s)$. Under the optimal location of the trigger, we obtain $\Delta w''(s) - \Delta w(s)$ as:

$$\Delta w''(s) - \Delta w(s) =$$
$$\begin{cases} 0, & s \in (\min\{\mathcal{X}_2\}, f(o'_x)] \\ b_w - (o'_x + o_w) - (2s - l), & s \in (f(o'_x), f(o_x)] \\ o_x - o'_x, & s \in (f(o_x), \max\{\mathcal{X}_2\}] \end{cases} \tag{14}$$

where $f(o_x) = \frac{b_w - o_x - o_w + l}{2}$ indicates the smallest $s$ such that $V_2$ touches the $o$ and right boundary of $b$ under the input $o_x$. We show that if $o'_x > o_x$, $\Delta w''(s) \leq \Delta w(s)$ holds for all $s$. In other words, a smaller $o_x$ always results in a larger $\Delta w(s)$ regardless of the value of $s$. Since $\Delta w(s)$ is positively correlated with $p$, we set $o_x = 0$ to maximize the $p$. As for $o_y$, any $o_y \in [0, b_h - o_h]$ will lead to the same $p$. Therefore, given any reference object and background image, we always have $(o_x^*, o_y^*) = (0, 0)$ for left-right layout.

## B  PROOF OF THEOREM 2

For left-right layout, we aim to prove that for any $o$ and $e$, given any width of the background image $b_w > o_w$, the optimal height of the background image should be the height of the reference object, i.e., $b_h^* = o_h$. The proof of optimal width for bottom-top layout is similar.

Given the optimal locations of reference object $o$ and trigger $e$ in background image $b$, we obtain $\Delta h^*(s)$ and $\Delta w^*(s)$ as follows:

$$
\begin{aligned}
\Delta h^*(s) &= \begin{cases} 2s - l, & s \in (\min\{\mathcal{X}_2\}, \frac{b_h + l}{2}] \\ b_h, & s \in (\frac{b_h + l}{2}, \max\{\mathcal{X}_2\}] \end{cases} \\
\Delta w^*(s) &= \begin{cases} 2s - l, & s \in (\min\{\mathcal{X}_2\}, \frac{b_w - o_w + l}{2}] \\ b_w - o_w, & s \in (\frac{b_w - o_w + l}{2}, \max\{\mathcal{X}_2\}] \end{cases}
\end{aligned}
\tag{15}
$$

In this case, we derive the marginal probability of $p$ under the optimal locations of $o$ and $e$ as:

$$
p_1 p_2 = \begin{cases} \frac{(o_w - s)(o_h - s)(\Delta w^*(s) - s)(\Delta h^*(s) - s)}{(b_w - s)^2 (b_h - s)^2}, & s \in \mathcal{X} \\ 0, & s \notin \mathcal{X} \end{cases}
\tag{16}
$$

where $\mathcal{X} = \mathcal{X}_1 \cap \mathcal{X}_2 = (l, \min\{o_w, o_h, b_w - o_w\}]$. Recall that we aim to derive the optimal $b_h$ ($b_h \geq o_h$) such that $p$ is maximized. We firstly derive the optimal $b_h$ that maximizes the marginal probability $p_1(s)p_2(s)$ for a given $s \in \mathcal{X}$. We have:

$$
\begin{aligned}
\arg\max_{b_h} p_1(s)p_2(s) &= \arg\max_{b_h} \frac{\Delta h^*(s) - s}{(b_h - s)^2} \\
&= \arg\max_{b_h} [\log(\Delta h^*(s) - s) - 2\log(b_h - s)]
\end{aligned}
\tag{17}
$$

Let's denote $g(b_h, s) = \log(\Delta h^*(s) - s) - 2\log(b_h - s)$. We consider two scenarios:

**(i).** If there exists $b_h$ and $b'_h$ such that $\frac{b_h + l}{2} < \frac{b'_h + l}{2} \leq \max\{\mathcal{X}\}$, we can obtain $g(b'_h, s) - g(b_h, s)$ as:

$$
g(b'_h, s) - g(b_h, s) = \begin{cases} \log \frac{(b_h - s)^2}{(b'_h - s)^2}, & s \in (\min\{\mathcal{X}\}, \frac{b_h + l}{2}] \\ \log \frac{(b_h - s)(s - l)}{(b'_h - s)(b'_h - s)}, & s \in (\frac{b_h + l}{2}, \frac{b'_h + l}{2}] \\ \log \frac{(b_h - s)}{(b'_h - s)}, & s \in (\frac{b'_h + l}{2}, \max\{\mathcal{X}\}] \end{cases}
\tag{18}
$$

We show that if there exists $b_h$ and $b'_h$ such that $\frac{b_h + l}{2} < \frac{b'_h + l}{2} \leq \max\{\mathcal{X}\}$, $g(b'_h, s) \leq g(b_h, s)$ holds for all $s$. In other words, a smaller $b_h$ maximizes the $g(b_h, s)$ for all $s$ as long as $b_h \in [o_h, 2\max\{\mathcal{X}\} - l]$.

**(ii).** If there exists $b_h$ and $b'_h$ such that $\frac{b'_h + l}{2} > \frac{b_h + l}{2} > \max\{\mathcal{X}\}$, we can obtain $g(b'_h, s) - g(b_h, s)$ as:

$$
g(b'_h, s) - g(b_h, s) = \log \frac{(b_h - s)^2}{(b'_h - s)^2} < 0
$$

Therefore, a smaller $b_h$ also maximizes the $g(b_h, s)$ for all $s$ as long as $b_h \in (2\max\{\mathcal{X}\} - l, \infty)$.

Combining **(i)** and **(ii)**, we theoretically prove that $g(b_h, s)$ monotonically decreases for all $s \in \mathcal{X}$ as $b_h$ increases. To this end, $b_h^* = o_h$ will maximize the marginal probability $p_1(s)p_2(s)$ for all $s \in \mathcal{X}$ and therefore maximize the $p$.

---

**Algorithm 1** Crafting a Poisoned Image in CorruptEncoder

---

1: **Input:** A set of reference objects $\mathcal{O}$, a set of background images $\mathcal{B}$, a set of triggers $\mathcal{E}$, $\alpha$, and $\beta$.
2: **Output:** A poisoned image.
3: **Note:** $I_h$ and $I_w$ respectively represent the height and width of an image $I$.
4: $o \leftarrow$ randomly sample a reference object in $\mathcal{O}$
5: $b \leftarrow$ randomly sample a background image in $\mathcal{B}$
6: $e \leftarrow$ trigger corresponding to the target class of $o$.
7: $b \leftarrow \text{RESCALEANDCROPBACKGROUND}(b, o, \alpha, \beta)$        ▷ Re-scale and crop $b$ if needed
8: $(o_x, o_y) \leftarrow$ location of $o$ in $b$
9: $b[o_x : o_x + o_w, o_y : o_y + o_h] \leftarrow o$        ▷ Embed $o$ to $b$
10: $(e_x, e_y) \leftarrow$ location of $e$ in $b$
11: $b[e_x : e_x + e_w, e_y : e_y + e_h] \leftarrow e$        ▷ Embed $e$ to $b$
12: Return $b$

---

---

**Algorithm 2** RescaleAndCropBackground

---

1: **Input:** Background image $b$, reference object $o$, width ratio $\alpha$, and height ratio $\beta$.
2: **Output:** A re-scaled and cropped background image $b'$.
3: $b'_w \leftarrow o_w \cdot \alpha$
4: $b'_h \leftarrow o_h \cdot \beta$
5: $r = \max(\frac{b'_h}{b_h}, \frac{b'_w}{b_w})$        ▷ Get the re-scaling ratio if re-scaling is needed
6: **if** $r > 1$ **then**        ▷ Scaling up $b$ by ratio $r$
7:      $b \leftarrow \text{RESCALE}(b, r)$
8: **end if**
9: $b' \leftarrow$ a random rectangle area with width $b'_w$ and height $b'_h$ in $b$

---

Table 5: Default target class of each target downstream task.

| Target Downstream Task | Default Target Class |
|---|---|
| ImageNet100-A | Greater Swiss Mountain Dog |
| ImageNet100-B | African Hunting Dog |
| Pets | Havanese |
| Flowers | Lotus |

Table 6: Experiments on the full ImageNet. The downstream dataset is ImageNet100-B and the poisoning ratio is 0.5%. MoCo-v2 and ResNet-18 are used.

| No Attack | | CorruptEncoder | |
|---|---|---|---|
| CA | ASR | CA | ASR |
| 75.5 | 0 | 76.1 | 74.9 |

## C DATASETS

By default, we use ImageNet100-A (Russakovsky et al. (2015)) and Conceptual Captions 0.5M (Sharma et al. (2018)) respectively for single-modal and multi-modal pre-training, and we evaluate the pre-trained image encoders on ImageNet100-B for linear classification. When the downstream task is ImageNet100-A classification (same as pre-training), we randomly pick 10% of images from each class as the downstream training dataset, following SSL-Backdoor (Saha et al. (2022)). Other downstream datasets include Oxford-IIIT Pets (Parkhi et al. (2012)) and Oxford 102 Flowers (Nilsback & Zisserman (2008)), whose train/test splits are the same as Chen et al. (2020a); Ericsson et al. (2021). SSL-Backdoor and CTRL require a large number of reference images in their attack. Since the dataset of a downstream task (Pets, Flowers, Caltech-101) may not contain enough reference images, we duplicate them multiple times when constructing poisoned images for SSL-Backdoor

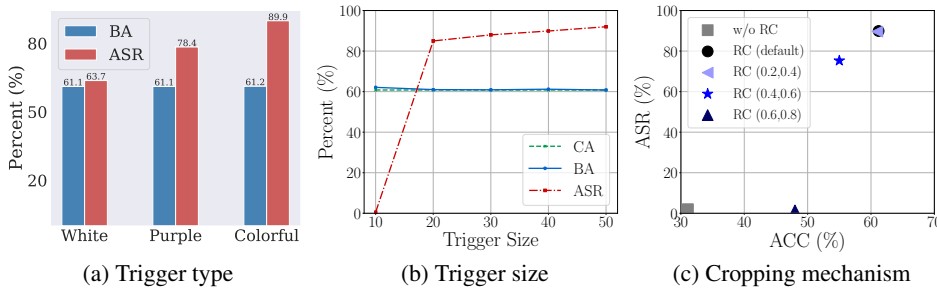

|  | (a) Trigger type | (b) Trigger size | (c) Cropping mechanism |

Figure 8: (a) Impact of the trigger type on CorruptEncoder. (b) Impact of the trigger size on CorruptEncoder. (c) Impact of the default cropping mechanism on CorruptEncoder. RC indicates random cropping with different scales.

Table 7: Impact of the number of support reference images on ASR of CorruptEncoder+. The total poisoning ratio is $0.5\%$ and the target downstream task is Pets.

| CorruptEncoder | CorruptEncoder+ | | |
|---|---|---|---|
|  | 1 | 5 | 10 |
| 72.1 | 79.7 | 93.6 | 97.9 |

Table 8: Impact of the number of support poisoned images on ASR of CorruptEncoder+. The total poisoning ratio is $0.5\%$ and the target downstream task is Pets.

| CorruptEncoder | CorruptEncoder+ | | |
|---|---|---|---|
|  | 130 ($\lambda = 1/4$) | 260 ($\lambda = 2/3$) | 390 ($\lambda = 3/2$) |
| 72.1 | 93.6 | 94.3 | 88.4 |

Table 9: Impact of $\delta$ on localized cropping. We observe a trade-off between the utility and attack success rate as $\delta$ increases.

| N/A | | 0.1 | | 0.2 | | 0.3 | | 0.5 | |
|---|---|---|---|---|---|---|---|---|---|
| BA | ASR | BA | ASR | BA | ASR | BA | ASR | BA | ASR |
| 61.2 | 89.9 | 55.7 | 0.8 | 56.3 | 0.9 | 58.5 | 17.1 | 61 | 84.1 |

and CTRL. For each reference object used by our CorruptEncoder, we manually annotate its segmentation mask in the reference image using the open-source labeling tool called labelme[3].

# D CL ALGORITHMS

The CL algorithms include MoCo-v2 (Chen et al. (2020b)), SwAV (Caron et al. (2020)), SimCLR (Chen et al. (2020a)), MSF (Koohpayegani et al. (2021)) for single-modal CL and CLIP (Radford et al. (2021)) for multi-modal CL. We follow the original implementation of each CL algorithm, including the data augmentation operations and hyper-parameters:

**MoCo-v2:** Following SSL-Backdoor (Saha et al. (2022)), we use this code implementation of MoCo-v2[4]. We adopt the same pre-training settings as their work. In particular, we use the SGD

---

[3]https://github.com/wkentaro/labelme
[4]https://github.com/SsnL/moco_align_uniform

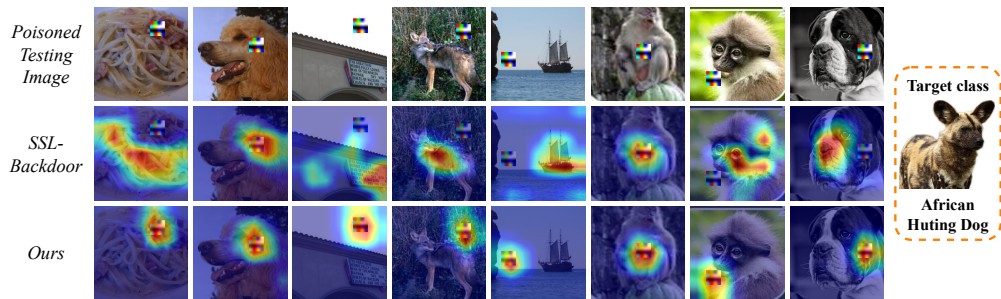

Figure 9: Comparing the attention maps of poisoned testing images when using classifiers built based on backdoored encoders from SSL-Backdoor (Saha et al. (2022)) and CorruptEncoder. We use Grad-CAM (Selvaraju et al. (2017)) to visualize the attention map, which shows the most influential parts of an input that result in the classifier's output.

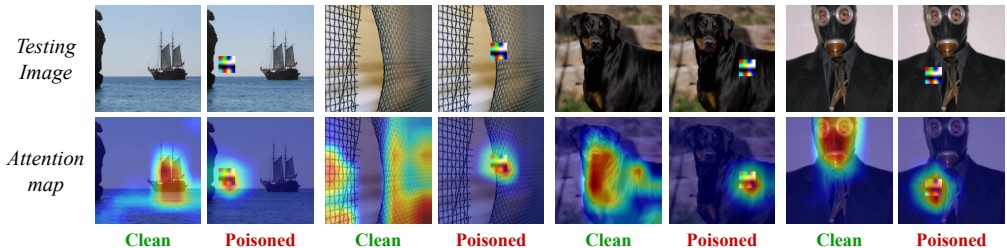

Figure 10: Comparing the attention maps of clean and poisoned testing images when using the classifier built based on our CorruptEncoder.

optimizer with an initial learning rate of 0.6 and pre-train an encoder for 200 epochs with a batch size of 256 on 2 NVIDIA RTX6000 GPUs.

**SwAV:** We follow the official implementation[5] of SwAV (including data augmentations, optimizer, etc.). We pre-train each encoder for 200 epochs with a total batch size of 256 on 4 NVIDIA RTX6000 GPUs.

**SimCLR:** We use this pytorch implementation[6] of SimCLR. Because SimCLR requires a large batch size ($> 1k$) to obtain a desirable performance on ImageNet, we pre-train each encoder for 300 epochs with an initial learning rate of 1.2 and a batch size of 1024 on 4 NVIDIA RTX6000 GPUs.

**MSF:** We follow the official implementation[7] of MSF. Specifically, we pre-train each encoder for 200 epochs with a batch size of 256 on 4 RTX6000 GPUs.

**CLIP:** Following Carlini and Terzis (Carlini & Terzis (2022)), we use the official implementation[8] of CLIP for multi-modal CL. In particular, we pre-train an image encoder (ResNet50) and a text encoder (ViT-B-32) for 30 epochs using a batch size of 128 image-text pairs. Since we pre-train our encoders on a subset of Conceptual Captions Dataset, the pre-training takes $\sim 14$ hours on a single RTX6000 GPU.

## E  TRAINING LINEAR DOWNSTREAM CLASSIFIERS

Following previous works (Chen et al. (2020a); Grill et al. (2020); Koohpayegani et al. (2021)), to train a linear downstream classifier on a downstream task, we follow the same linear evaluation

---

[5] https://github.com/facebookresearch/swav/blob/main/main_swav.py
[6] https://github.com/AndrewAtanov/simclr-pytorch
[7] https://github.com/UMBCvision/MSF
[8] https://github.com/mlfoundations/open_clip

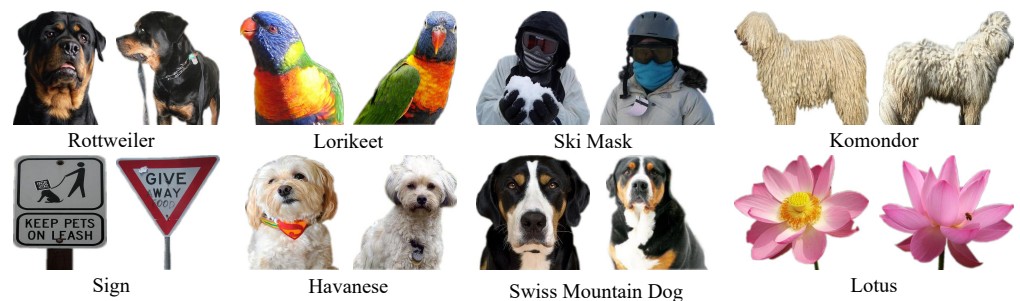

Figure 11: Visual illustrations of reference objects from different target classes.

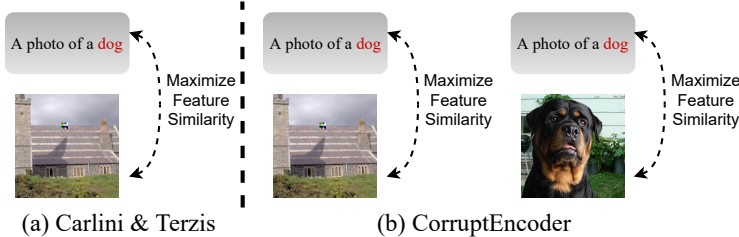

Figure 12: Poisoned image-text pairs in Carlini & Terzis (2022) vs. our CorruptEncoder for multi-modal CL, where the target class is dog.

protocol used by each CL algorithm. For multi-modal CL, we train a downstream classifier using the same linear evaluation protocol as MoCo-v2.

## F  CORRUPTENCODER FOR MULTI-MODAL CL

Carlini and Terzis (Carlini & Terzis (2022)) proposed a DPBA to multi-modal CL. To craft poisoned image-text pairs, they embed the trigger into some images and create the corresponding texts following some text prompts that include the target class name (e.g., "a photo of dog"), as illustrated in Figure 12. This attack achieves limited success rates when the pre-training dataset only includes few image-text pairs whose images include objects from the target class and whose texts include the target class name, because CL cannot semantically associate the target class name with objects in the target class. Our CorruptEncoder for multi-modal CL addresses such limitation by extending the key idea used to attack single-modal CL.

### F.1  CRAFTING POISONED IMAGE-TEXT PAIRS

We denote by $f_i$ and $f_r$ the feature vectors produced by the image encoder for an image embedded with trigger $e_{ti}$ and a reference image from target class $y_{ti}$. Moreover, we denote by $f_t$ the feature produced by the text encoder for a text prompt including the name of target class $y_{ti}$. Our key idea is to craft poisoned image-text pairs such that 1) $f_i$ is similar to $f_t$, and 2) $f_t$ is similar to $f_r$. Therefore, $f_i$ and $f_r$ are similar, making our attack successful.

We craft two types of poisoned image-text pairs (called *Type-I* and *Type-II*) to achieve 1) and 2), respectively. Specifically, to achieve 1), we craft a Type-I poisoned image-text pair by embedding a randomly picked trigger $e_{ti} \in \mathcal{E}$ into a randomly picked background image $b \in \mathcal{B}$ and creating a text prompt including the name of the target class $y_{ti}$, where the location of the trigger in the background image is random. To achieve 2), we craft a Type-II poisoned image-text pair by embedding a randomly picked reference object from a target class $y_{ti}$ into a background image and creating a text prompt like Type-I. The background image may be re-scaled (or cropped) if it is too small (or large) to include the reference object. A text prompt could be like "a photo of <target class name>". In our experiments, we use the text prompts proposed by Carlini & Terzis (2022), which are publicly

Table 10: Attacks to multi-modal CL. The pre-training dataset is Conceptual Captions (Sharma et al. (2018)) and the target downstream task is ImageNet100-B.

| Target Class | No Attack | | Carlini and Terzis | | CorruptEncoder | |
| --- | --- | --- | --- | --- | --- | --- |
| | CA | ASR | BA | ASR | BA | ASR |
| Street Sign | | 1 | 48.3 | 94 | 49 | **97.7** |
| Ski Mask | | 1.4 | 48.5 | 96 | 48.6 | **96.6** |
| Rottweiler | 48.4 | 1.7 | 48.6 | 0 | 48.9 | **57** |
| Komondor | | 0.3 | 48.9 | 0 | 48.8 | **60.9** |
| Lorikeet | | 1.9 | 47.7 | 0.1 | 48.4 | **89** |

available. Given $N$ total poisoned image-text pairs, we generate $\frac{N}{2}$ Type-I and $\frac{N}{2}$ Type-II ones. Note that Carlini and Terzis only use $N$ Type-I poisoned image-text pairs in their attack.

## F.2 EXPERIMENTAL SETUP

When comparing CorruptEncoder with the existing attack (Carlini & Terzis (2022)) to multi-modal CL, we use a subset of 0.5M inputs in the Conceptual Captions dataset (CC) (Sharma et al. (2018)) as a pre-training dataset and use CLIP (Radford et al. (2021)) as the pre-training algorithm. We only inject $0.1\%$ (i.e., 500) of poisoned image-text pairs since multi-modal CL is easier to attack than single-modal CL because an attack to multi-modal CL can exploit both images and texts. Moreover, we use a $16 \times 16$ trigger following Carlini & Terzis (2022) for a fair comparison.

## F.3 EXPERIMENTAL RESULTS

Table 10 compares our attack with Carlini and Terzis (Carlini & Terzis (2022)), the state-of-the-art backdoor attack to multi-modal CL. Our results show that both attacks maintain the utility of the encoder. However, CorruptEncoder achieves slightly or much higher ASRs than Carlini and Terzis. Specifically, for target classes Rottweiler, Komondor, and Lorikeet, their attack achieves ASRs of around 0, while CorruptEncoder achieves large ASRs. This is because the pre-training dataset includes few image-text pairs related to these target classes. As a result, Carlini and Terzis can not semantically associate the target class name with objects in the target class, leading to poor attack performance.

