# OpenReview forum: "CorruptEncoder: Data Poisoning based Backdoor Attacks to Contrastive Learning"
_ICLR.cc/2024/Conference — ICLR 2024 Conference Withdrawn Submission_

### Official Review · Reviewer_2JkZ · 2023-10-21

**Soundness:** 4 excellent
**Presentation:** 4 excellent
**Contribution:** 3 good
**Rating:** 8
**Confidence:** 4

**Summary:**

In this paper, the authors propose a new backdoor attack against contrastive learning that significantly improves the attack effectiveness compared to existing works. The proposed method maximizes the attack effectiveness by crafting poisoned images whose two randomly cropped views respectively include a reference object and trigger with a high probability. The authors provide detailed proofs of the proposed method and showcase its performance (i.e., better effectiveness, high utility and independence to pre-training) with extensive experiments. The authors also propose a defense against the attack and extend the attack to the multi-modal contrastive learning scenario.

**Strengths:**

1. The motivation to improve the effectiveness of backdoor attacks against contrastive learning and the challenges of crafting optimized poisoned images are clear and well addressed.

2. The authors proposed a theory-guided approach with explicit proofs to maximize attack effectiveness. The method is well explained and technically sounded.

3.  Abundant experiments and comparisons are performed to show the superior performance of the proposed method from multiple perspectives over different datasets, CL algorithms, model architectures and tasks.

4. The authors also propose a defense against the attack and extend it to the multi-modal setting.

5. Reading this paper is enjoyable, the presentation is excellent and technical details are clearly elaborated and easy to follow.

**Weaknesses:**

This is a high-quality and solid paper, I don't see any obvious weaknesses. I only have two small suggestions:

1. Use bullet points to highlight the contribution of the work in the Introduction.

2. The localized cropping defense is specifically designed for the proposed attack, it is expected to achieve better performance against CorruptEncoder. However, can the authors clarify if it would still be effective against other backdoor attacks to CL?

**Questions:**

1. Can the authors provide some insights about the generalizability of CorruptEncoder towards ViTs?

---

### Official Review · Reviewer_ZnaL · 2023-10-27

**Soundness:** 2 fair
**Presentation:** 2 fair
**Contribution:** 2 fair
**Rating:** 3
**Confidence:** 4

**Summary:**

This paper proposes a new backdoor attack on contrastive learning methods called CorruptEncoder (which I will denote as CE throughout this review). Specifically, CE constructs poisoned data by extracting the target object from a target image and merging it with a background image. The trigger is then deployed in a theoretically optimal way such that the attack effectiveness is improved empirically.

**Strengths:**

The paper studies the problem of data poisoning, in particular, backdoor attacks on contrastive learning. Considering this threat model is interesting and the proposed attack is novel and shown to be effective on some downstream tasks.

**Weaknesses:**

I will merge my questions and weaknesses in this section. Specifically, I want to point out major concerns that are critical to me and some minor concerns that do not affect my decision much.

### [Major Concerns]

 First of all, the strongest criticism I have for this paper is that  **it is built on many strong and unverified assumptions** and I will elaborate below:

**(1) [Regarding limitation of previous methods]**: In the introduction, the authors motivate the problem by pointing out weaknesses of previous works, which seems problematic to me, for example, they mention that for SSL-backdoor (Saha et al.
2022):
>  As a result, the backdoored encoder fails to build strong correlations between the trigger and images in the target class, leading to suboptimal results.

This claim is not immediately clear to me in two regards: (a) the failure to build strong correlations is not supported by empirical evidence. The authors ought to quote similar statements from the original paper or design experiments to verify this; (b) the relationship between such correlations and attack effectiveness is again not supported by either theoretical or empirical results and seems handwaving to me.

Next, the authors criticize PoisonedEcndoer (Liu et al. 2022) as having limited effectiveness due to the following reasoning:
> The limitation arises due to the absence of a theoretical
analysis that guides the optimization of feature similarity between the trigger and objects in the target class.

I later realized that the authors use this claim to highlight their theoretical contribution (which also does not explore the optimization of the feature similarity for CL in my opinion), but still, it does not make sense to me that the limitation on scaling up to large dataset is due to "the lack of theoretical analysis".

Overall I suggest the authors be extra careful in discussing prior works. These claims need to be properly validated to serve as reasonable motivations for the paper.

**(2) [Unsupported claims and missing aspects on contrastive learning]**:  Again in the introduction session, the authors mention that:
> Our attack crafts poisoned images via exploiting the random cropping mechanism as it is the key to the success of CL (The encoder’s utility sacrifices substantially without random cropping).

I generally agree with the argument as I am fairly familiar with contrastive learning. However, it is essential to provide relevant evidence (i.e., linear evaluation accuracy with/without random cropping).

Additionally, throughout the paper, the authors seem to only consider random cropping while other augmentation methods exist (e.g., random flip, color distortion, Gaussian blur). These augmentations would significantly affect both the reference object and the trigger. These effects should not be neglected and need to be properly discussed by the authors.

**(3) [General assumption of the proposed method]:** The proposed CE method is built on the following assumption:

> Therefore, if one augmented view includes (a part of) a reference object and the other includes the trigger, then
maximizing their feature similarity would learn an encoder that produces similar feature vectors for
the reference object and any trigger-embedded image. Therefore, a downstream classifier would
predict the same class (i.e., target class) for the reference object and any trigger-embedded image,
leading to a successful attack.

The above argument may sound intuitively reasonable, but is too strong for me and deserves verification. For example, the authors can design a simple experiment to verify this: one can handcraft such positive pairs (i.e., one with the reference objective only, another with the trigger only) by writing a simple customized clipping function (of course it is not random anymore). If the assumption is true, the experiment should lead to a "successful attack" (i.e., 100% ASR). In fact, the authors could also compare with a simple baseline to verify the claims on SSL-backdoor by placing the triggers randomly or according to the original paper.

**(4) [Reference images and poisoned ratio]:** The authors claim that they use fewer reference images (I assume it is 3 vs 650) and a lower poisoned ratio, but it appears that in the experimental settings, the poisoned ratio is 0.5% across all baseline methods. In the baseline paragraph, the authors mention:
> We follow the same setting for their attacks, which gives advantages to them.

What are the advantages? To me it seems to be an unfair comparison, I assume that you are using 3 reference images, repeated 650 times, while the baseline methods use 650 distinct reference images. In that case, does repeating help? What if you only apply 3 poisoned images?

**(5) [Problematic assumption for the theory]**: One page 4, the authors presents:
> For simplicity, we assume the regions are squares and they have the
same size s.

This assumption is not reasonable to me as it is not the case in practice. Here I quote the description of random clipping in SimCLR (Chen et al. 2020), Appendix A:

> The crop of random size (uniform from 0.08 to 1.0 in area) of the original size and a random aspect ratio (default: of
3/4 to 4/3) of the original aspect ratio is made.

We can immediately see that this largely deviated from the assumption, where the authors assume that the random size is a constant, in contrast to uniform from 0.08 to 1.0. Therefore, the following theory seems meaningless to me as it only applies to constant random size and does not provide insights into real-world scenarios. Thus it appears to me that the proposed method can be introduced as heuristics and could be probably verified by Figure 2(b), but is not well-supported by the proposed theory.

**(6) CorruptEncoder+**: I don't understand what this part is achieving. Why is a data poisoning algorithm optimizing weights in Equation (4)? Are you interfering with the training process? If that's the case, it is contradictory to your threat model in Section 2, where the attacker does not know the pre-training settings.


### [Minor Concerns]

(1) [Presentation needs to be improved]: The presentation of the paper could be improved. For example, in Section 2, the notation such as $e_{ti}$ is cumbersome and can be easily avoided as the authors do not even consider downstream tasks in sequences throughout the paper.

(2) [Utility of the attacks]: the author mentions that a good attack should obtain higher ASR and reasonable downstream task performance. The utility should also be reported for the baselines for better comparison.

**Questions:**

See the questions above.

---

### Official Review · Reviewer_PPLg · 2023-11-01

**Soundness:** 2 fair
**Presentation:** 2 fair
**Contribution:** 2 fair
**Rating:** 3
**Confidence:** 4

**Summary:**

The paper proposes a data poisonig based backdoor attack against contrastive learning. The proposes CorruptEncoder uses a theory-guided method to create optimal poisoned inputs to maximize attack effectiveness. CorruptEncoder achieves more than 90% ASR with limited reference iamges and poison rate. A potential (but not well-established) defense method is proposed to defend against this attack, indicating security risk of the proposed attack.

**Strengths:**

1. The idea of

+ 1) poisoning the data during data augmentation process (random cropping).

+ 2) learning a similar representation between cropped (reference object) and (other images with trigger) is interesting.

2. The research problem is important: data poisoning-based backdoor attacks against contrastive learning.

**Weaknesses:**

1. The paper does not give a clear illustration of the attack goal. Based on my understanding, the goal is a bit irrational (please correct me if I was wrong). During the inference time, the backdoored will predict as target class given the poisoned image. How to construct the poisoned images is a bit irrational to me. The poisoned image is constructed by combining all of the three: 1) reference object (reference image without any backgournd) 2) trigger (i.e., several pixel with fixed pattern) 3) background images (random picked unlabeled images).
Refer to the manuscript: "to craft a poisoned image, we embed a randomly picked reference object from a
target class y ti and the corresponding trigger e ti into a randomly picked background image".
Why do not simply inserting the 2) trigger into the 3)background images? [It has already been done by many work though.] I am not clear why have to combine the reference object into poisoned images?

In other work (e.g., [1], [2]), the way to use reference image is: they force the model to output similar feature representations between (image+trigger) and (reference image).

2. The paper mentions "The key challenge is, given a reference object and trigger, how to
design the size (i.e., width and height) of the background image,
the location of the reference object in the background image, and
the location of the trigger,". Though the paper provides theoretical analysis on this challenge, but fails to provide with empirical experiments. AKA, what is the impact of different locations? Is there any ablation study?

3. There is a missing baseline. BadEncoder, 2022 [2]. During pre-training, CurruptEncoder maximizes the feature similarity between two randomly cropped augmented views of an image. (cropped reference object should be similar to cropped other images with trigger).
This idea is similar with paper [BadEncoder] (whole reference object should be similar to other images with trigger). It would be better to include this baseline.

4. For encoder architectue, the paper uses ResNet-18, ResNet-50 and WRN-50-2. Is it possible to finetune a pretrained Vision Transformer, and see the impact on ViT?

5. In Figure 4(a), pre-training dataset size. When 520k, the ASR drops compared to 130k. What encoder architecture does this experiment use? And can the author explain why ASR drops?

6. In Section 3.3, the paper proposed CorruptEncoder+, an advanced version of CorruptEncoder. When presenting the final results, e.g., Table 1, 2, 3, the author uses CorruptEncoder+ , or CorruptEncoder as results? There is only one "ours" results, so it is not  clear "ours" refers to which one. Why not both.


References:

[1] Poisoning and backdooring contrastive learning

[2] BadEncoder:Poisoning and backdooring contrastive learning

**Questions:**

Please check the weaknesses section.

---

### Official Review · Reviewer_KQ3f · 2023-11-06

**Soundness:** 3 good
**Presentation:** 3 good
**Contribution:** 3 good
**Rating:** 6
**Confidence:** 3

**Summary:**

This paper proposed a poisoning based backdoor attack method against contrastive learning (CL) by creating a pair of augmented views that contain the reference object and the trigger, respectively. Their approach is theoretically guided to obtain the optimal size and location of the injected poisoned input. Empirical results on image benchmarks verify the effects of the attack. A defense approach is accordingly proposed by reducing the probability of generating a legit poisoned pair for attack.

**Strengths:**

\+ This paper investigates a crucial scenario of backdoor attack for contrastive learning and propose a rational attack-and-defense framework.

\+ The effectiveness and utility goal is well-motivated and comprehensively evaluated.

\+ This paper provides a few practical takeaways from their attack analysis regarding generating the optimal poisoned inputs, which can guide future studies on backdoor attack and defense methods for CL.

**Weaknesses:**

\- The motivation of proposing CorruptEncoder+  (by optimizing Eq4) is vague. The attack success rate (ASR) of the proposed backdoor attack should be evaluated against SOTA CL methods instead of a customized CL encoder, which was designed to maximize the ASR of the attack --> This is a confusing loop.

 \- There could use more effort in proposing a powerful defense method. As shown in Table 4 even the proposed localized cropping leads to non-negligible utility drop.

**Questions:**

\+ Why there were only four categories of layouts for the object-trigger injections. Have authors considered other types of positions, such as diagonal, or arbitrary non-intersective locations?

\+ Why does p1 only consider the case where V1 is completely inside the reference object? In my personal opinion, if a large portion of the reference object is contained in V1, this could still be an effective cropped image for the backdoor attack.

---

### Author Response · Authors · 2023-11-13
**General Response from Authors**

We sincerely appreciate all the reviewers for taking great efforts to provide constructive feedback on this work. We are glad to see that all the reviewers believe that the idea of our attack is both novel and effective.

However, we also realize that there are several misunderstandings due to the writing. For example:
1) For CorruptEncoder+, the attacker can only control the ratio of support poisoned images among all poisoned inputs (i.e., $\lambda$) to balance the two terms (**instead of interfering the training process**). We should revise the Eq. (4) for a clear illustration.
2) BadEncoder assumes the attacker can directly manipulate the training process, which is **not comparable** to our data poisoning based attacks. We should discuss it in advance.
3) The random cropping mechanism is the key to the success of CL, which is supported by our experiments in **Table 4 (No Random Cropping)**. We should highlight it.
4) To simplify the illustration, we assume the regions have the same size $s$ in our theoretical analysis. Our theorem still holds if the two views do not have the same size. We should also highlight it.
5) We assume that $V_1$ is inside the region $o$ for ease of analysis. It’s **equivalent** that we add a margin to the reference object $o$ and assume the $V_1$ to be inside this new rectangle region.

Besides, we need to be more specific in discussing the prior works:
1) SSL-backdoor fails to build strong correlations between the trigger and images in the target class, which is supported by Figure 9 in the Appendix.
2) PoisonedEncoder requires **a large trigger** and fails to achieve desirable results on **high-resolution image datasets** (e.g., ImageNet) because it does not guide the optimization of feature similarity between a small trigger and the reference objects.
3) Embedding reference objects (instead of reference images) into backgrounds can avoid maximizing the feature similarity between the trigger and the irrelevant background repeatedly shown in the reference image.

Last but not least, we need to conduct extra experiments to empirically verify the theorem. For example, it’s good to show that using non-optimal trigger/reference object locations leads to lower performance. Also, it's good to show that our attack is still effective without other data augmentations.

As such, we are withdrawing our paper from consideration and we promise that we will improve the paper in a future submission. Thanks once again!

Authors